# How to Backdoor HyperNetwork in Personalized Federated Learning?

## Abstract

This paper explores previously unknown backdoor risks in HyperNet-based personalized federated learning (HyperNetFL) through poisoning attacks. Based upon that, we propose a novel model transferring attack (called HNTROJ), i.e., the first of its kind, to transfer a local backdoor infected model to all legitimate and personalized local models, which are generated by the HyperNetFL model, through consistent and effective malicious local gradients computed across all compromised clients in the whole training process. As a result, HNTROJ reduces the number of compromised clients needed to successfully launch the attack without any observable signs of sudden shifts or degradation regarding model utility on legitimate data samples making our attack stealthy. To defend against HNTROJ, we adapted several backdoor-resistant FL training algorithms into HyperNetFL. An extensive experiment that is carried out using several benchmark datasets shows that HNTROJ significantly outperforms data poisoning and model replacement attacks and bypasses robust training algorithms even with modest numbers of compromised clients.

## 1 Introduction

Recent data privacy regulations [1] pose significant challenges for machine learning (ML) applications that collect sensitive user data. Federated learning (FL) [2] is a promising way to address these challenges by enabling clients to jointly train ML models via a coordinating server without sharing their data. Although offering better data privacy, FL typically suffers from the disparity of model performance caused by the non-independent and identically distributed (non-iid) data distribution across clients [3]. One of the state-of-the-art approaches to address this problem in FL is using a single joint neural network, called HyperNetFL, to generate local models using personalized descriptors optimized for each client independently [4]. This allows us to perform smart gradient and parameter sharing.

Despite the superior performance, the unique training approach of HyperNetFL poses previously unknown risks to backdoor attacks typically carried out through poisoning in FL [5]. In backdoor attacks, an adversary manipulates the training process to cause model misclassification on a subset of chosen data samples [6,7]. In FL, the adversary tries to construct malicious gradients or model updates that encode the backdoor. When aggregated with other clients' updates, the aggregated model exhibits the backdoor.

We investigate backdoor attacks against HyperNetFL and formulate robust HyperNetFL training as defenses. Our developed attack (called HNTROJ) is based on consistently and effectively crafting malicious local gradients across compromised clients using a single backdoor-infected model to enforce HyperNetFL generating local backdoor-infected models disregarding their personalized descriptors. An extensive analysis and evaluation in non-iid settings show that HNTROJ notably outperforms existing model replacement and data poisoning attacks bypassing recently developed robust federated training algorithms adapted to HyperNetFL with small numbers of compromised clients. Therefore, it is challenging to defend HNTROJ.

Preprint. Under review.

## 2 Background

This section briefly reviews HyperNetFL and backdoor and poisoning attacks. More information is in Appx. A.

**HyperNetwork-based Personalized FL (HyperNetFL).** To address the disparity of model utility across clients, HyperNetFL [4, 8] uses a neural network $h(v_i, \varphi)$ located at the server to output the weights $\theta_i$ for each client $i$ using a (trainable) descriptor $v_i$ as input and model weights $\varphi$, that is, $\theta_i = h(v_i, \varphi)$. HyperNetFL offers a natural way to share information across clients through the weights $\varphi$ while maintaining the personalization of each client via the descriptor $v_i$. To achieve this goal, the clients and the server will try to minimize their loss functions: $\arg\min_{\varphi, \{v_i\}_{i \in [N]}} \frac{1}{N} \sum_{i \in [N]} L_i(h(v_i, \varphi))$.

**Backdoor and Poisoning Attacks.** Training time poisoning attacks against ML and FL models can be classified into byzantine and backdoor attacks. In byzantine attacks, the adversarial goal is to degrade or severely damage the model test accuracy [9–11]. Byzantine attacks are relatively detectable by tracking the model accuracy on validation data [12]. Meanwhile, in backdoor attacks, the adversarial goal is to cause model misclassification on a set of chosen inputs without affecting model accuracy on legitimate data samples. A well-known way to carry out backdoor attacks is using Trojans [13, 14]. A Trojan is a carefully crafted pattern, e.g., a brand logo, blank pixels, added into legitimate samples causing the desired misclassification. A recently developed image warping-based Trojan mildly deforms an image by applying a geometric transformation [15] to make it unnoticeable to humans and bypass all well-known Trojan detection methods, such as Neural Cleanse [16], Fine-Pruning [17], and STRIP [18]. The adversary applies the Trojan on legitimate data samples to activate the backdoor at the inference time.

The training data is scattered across clients in FL, and the server only observes local gradients. Therefore, backdoor attacks are typically carried by a small set of compromised clients fully controlled by an adversary to construct malicious local gradients and send them to the server. The adversary can apply data poisoning (DPOIS) and model replacement approaches to create malicious local gradients. In DPOIS [19, 20], compromised clients train their local models on Trojaned datasets to construct malicious local gradients, such that the aggregated model at the server exhibits the backdoor. DPOIS may take many training rounds to implant the backdoor into the aggregated model. Meanwhile, in model replacement [6], the adversary constructs malicious local gradients, such that the aggregated model at the server will closely approximate or be replaced by a predefined Trojaned model. To some extent, model replacement is highly severe since it can be effective after only one training round.

To our knowledge, *these attacks are not primarily designed for HyperNetFL, in which there is no aggregated model $\theta$ at the server.* That poses an unknown risk of backdoors through poisoning attacks in HyperNetFL.

## 3 Model Transferring Attack

To overcome the lack of consistency in deriving the malicious local gradients in DPOIS and avoid sudden shifts in model utility on legitimate data samples in HNREPL, we propose in this work a novel *model transferring attack* (HNTROJ) against HyperNetFL.

In HNTROJ (Alg. 1), our idea is to replace $\theta_t^c$ with a Trojaned model $X$ across all the compromised clients $c \in \mathtt{C}$ and in all communication rounds $t$ to compute the malicious local gradients: $\forall c \in \mathtt{C}, t \in [T] : \triangle \bar{\theta}_t^c = \psi_t^c [X - h(v_c, \varphi)]$, where $\psi_t^c$ is a dynamic learning rate randomly sampled following a specific distribution, e.g., uniform distribution $\mathcal{U}[a, b]$, $a < b$, and $a, b \in (0, 1]$. In practice, the adversary can collect its own data to locally train $X$. The collected data can be uniformly distributed across all classes to maximize the backdoor successful rate across all legitimate clients, whose data distribution is non-iid. This is because the server does not know the clients' local data distribution.

By doing so, we achieve several key advantages, as follows:

**(1)** The gradients $\triangle \bar{\theta}_t^c$ become more effective in creating backdoors, since $X$ is a better optimized Trojaned model than $\{\theta_t^{c*}\}_{c \in \mathtt{C}}$.

**(2)** The gradients across compromised clients synergistically approximate the Trojaned surrogate loss (Eq. 9, Appx. B) to closely align the outputs of $h(\cdot, \varphi)$ to a unified Trojan model $X$ through the term

**Algorithm 1** Model Transferring Attack in HyperNetFL (HNTROJ)

**Input:** Number of global rounds $T$ and local rounds $K$, learning rates $\lambda$, $\zeta$, and $\eta$, number of clients $N$, dynamic learning rate $\psi \sim \mathcal{U}[a,b]$, and $L_i(B)$ is the loss function $L_i(\theta)$ on a mini-batch $B$

**Output:** $\varphi, v_i$

1: **for** $t = 1, \ldots, T$ **do**
2:     Sample clients $S_t$
3:     **for** each legitimate client $i \in S_t \setminus \mathsf{C}$ **do**
4:        set $\theta_t^i = h(v_i, \varphi)$ and $\tilde{\theta}^i = \theta_t^i$
5:        **for** $k = 1, \ldots, K$ **do**
6:           sample mini-batch $B \subset D_i$
7:           $\tilde{\theta}_{k+1}^i = \tilde{\theta}_k^i - \eta \nabla_{\tilde{\theta}_k^i} L_i(B)$
8:        $\triangle \theta_t^i = \tilde{\theta}_K^i - \theta_t^i$
9:     **for** each compromised client $c \in S_t \cap \mathsf{C}$ **do**
10:       $\triangle \bar{\theta}_t^c = \left(\psi_t^c \sim \mathcal{U}[a,b]\right)\left[X - h(v_c, \varphi)\right]$
11:     $\varphi = \varphi - \frac{\lambda}{|S_t|} \sum_{i=1}^{|S_t|} (\nabla_\varphi \theta_t^i)^\top \triangle \theta_t^i$
12:     $\forall i \in S_t : v_i = v_i - \zeta \nabla_{v_i} \varphi^\top (\nabla_\varphi \theta_t^i)^\top \triangle \theta_t^i$

$\frac{1}{2} \sum_{c \in \mathsf{C}} \|X - h(v_c, \varphi)\|_2^2$ disregarding the varying descriptors $\{v_i\}_{i \in N}$ and their dissimilar local datasets. The new Trojaned surrogate loss is: $\frac{1}{2}\left(\sum_{c \in \mathsf{C}} \|X - h(v_c, \varphi)\|_2^2 + \sum_{i \in N \setminus \mathsf{C}} \|\theta^{i*} - h(v_i, \varphi)\|_2^2\right)$.

**(3)** The gradients $\triangle \bar{\theta}_t^c$ become stealthier since updating the HyperNetFL with $\triangle \bar{\theta}_t^c$ will significantly improve the model utility on legitimate data samples. This is because $X$ has a better model utility on legitimate data samples than the local models of legitimate clients $\{\theta_t^{i*}\}_{i \in N \setminus \mathsf{C}}$. More importantly, *by keeping the random and dynamic learning rate $\psi_t^c$ only known to the compromised client $c$*, we can prevent the server from tracking $X$ or identifying some suspicious behavior patterns from the compromised client.

We also quantify the server's estimation error $Error$ as follows.

Assuming that the server can identify compromised clients with a precision value $p$, we can quantify the server's estimation error bounds of the Trojaned model $X$, as follows. The server's set of identified compromised clients consists of $p|\mathsf{C}|$ compromised clients $\bar{\mathsf{C}}$ and $(1-p)|\mathsf{C}|$ legitimate clients $\bar{L}$. The estimated Trojaned model $X' = \sum_{c \in \bar{\mathsf{C}} \cup \bar{L}} \theta_t^c / |\mathsf{C}|$. Then, the estimation error of $X$ is computed and bounded as follows:

$$Error = \|\sum_{c \in \bar{\mathsf{C}}} \frac{\theta_t^c}{p|C|} + \sum_{i \in \bar{L}} \frac{\theta_t^i}{(1-p)(N-|C|)} - X\|_2$$

$$= \|X' - X\|_2 = \|\sum_{c \in \bar{\mathsf{C}} \cup \bar{L}} \frac{\theta_t^c}{|\mathsf{C}|} - X\|_2 \tag{1}$$

in which

$$\|X' - X\|_2 \geq \|\sum_{c \in \bar{\mathsf{C}}} \theta_t^c / (p|\mathsf{C}|) - X\|_2 = \|\sum_{c \in \bar{\mathsf{C}}} \frac{\triangle \bar{\theta}_t^c}{p|\mathsf{C}|\psi_t^c}\|_2 \geq \|\sum_{c \in \bar{\mathsf{C}}} \frac{\triangle \bar{\theta}_t^c}{p|\mathsf{C}|b}\|_2 \tag{2}$$

and

$$\|\sum_{c \in \bar{\mathsf{C}} \cup \bar{L}} \theta_t^c / |\mathsf{C}| - X\|_2 \leq \arg \max_{L \subseteq N \text{ s.t. } |L| = |\mathsf{C}|} \|\sum_{i \in L} \theta_t^i / |L| - X\|_2 \tag{3}$$

From Eqs. 2 and 3, we have the following error bounds:

$$\|\sum_{c \in \bar{\mathsf{C}}} \frac{\triangle \bar{\theta}_t^c}{p|\mathsf{C}|b}\|_2 \leq Error \leq \arg \max_{L \subseteq N \text{ s.t. } |L| = |\mathsf{C}|} \|\sum_{i \in L} \frac{\theta_t^i}{|L|} - X\|_2 \tag{4}$$

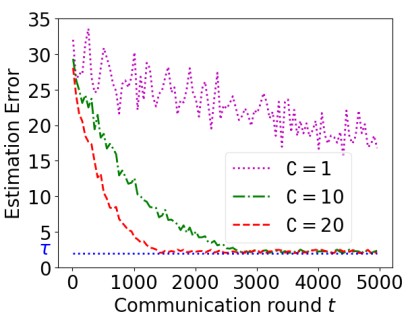

Figure 1: Estimation error of HNTROJ with $p = 1$.

We observe that: 1) The lower precision in detecting compromised clients (smaller $p$) results in a larger $Error$ approaching the upper bound; 2) The smaller $\psi_t^c$, the higher lower bound of $Error$ is; and 3) If the gradient $\triangle \bar{\theta}_t^c$ is too small, we can uniformly upscale its $L_2$-norm to be a small constant (denoted $\tau$) to enlarge the lower bound of $Error$ without affecting the model utility or backdoor success rate. Fig. 1 illustrates the lower bound of $Error$ given the most favorable precision to the server ($p = 1$) with different values of $\complement$. After 1,000 rounds, $Error$ is not further reduced since it is controlled by the lower bound (with $\tau = 2$ in this study). That prevents the server from accurately estimating $X$.

**(4)** The following Theorem 1 shows that the $L_2$-norm distance between the local model $\theta_t^c$ of a compromised client generated by the HyperNetFL $h(v_c, \varphi)$ and $X$, i.e., $\|\theta_t^c - X\|_2$, is bounded by $(\frac{1}{a} - 1)\| \triangle \bar{\theta}_{t'}^c \|_2 + \|\xi\|_2$, where $t'$ is the closest round the compromised client $c$ participated in before $t$, and $\xi \in \mathcal{R}^m$ is a small error rate. When the HyperNetFL model converges, e.g., $t', t \approx T$, $\|\xi\|_2$ become tiny and $\| \triangle \bar{\theta}_{t'}^c \|_2$ is bounded by a small constant $\tau$ ensuring that given the compromised client, $\theta_T^c = h(v_c, \varphi)$ converges into a bounded and low loss area surrounding $X$ ($\|\theta_T^c - X\|_2$ is tiny) to imitate the normal training process.

Consequently, HNTROJ requires a smaller number of compromised clients to be highly effective. Also, HNTROJ is stealthier than the (white-box) HNREPL and DPOIS by avoiding degradation and shifts in model utility on legitimate data samples during the whole poisoning process.

**Theorem 1.** *For a compromised client $c$ participating in a round $t \in [T]$, we have that the $L_2$-norm distance between the HyperNetFL output $\theta_t^c = h(v_c, \varphi)$ and the Trojaned model $X$ is always bounded as follows:*

$$\|\theta_t^c - X\|_2 \le (1/a - 1)\| \triangle \bar{\theta}_{t'}^c \|_2 + \|\xi\|_2 \tag{5}$$

*where $\forall t : \psi_t \sim \mathcal{U}[a, b]$, $a < b$, $a, b \in (0, 1]$, $t'$ is the closest round the compromised client $c$ participated in, and $\xi \in \mathcal{R}^m$ is a small error rate.*

*Proof.* At the round $t'$, we have that $\triangle \bar{\theta}_{t'}^c = \psi_{t'}^c [X - \theta_{t'}^c]$. This is equivalent to $X = \frac{\triangle \bar{\theta}_{t'}^c}{\psi_{t'}^c} + \theta_{t'}^c$. At the round $t$, the HyperNetFL $h(v_c, \varphi)$ supposes to generate a better local model for the compromised client $c$: $\theta_t^c = \triangle \bar{\theta}_{t'}^c + \theta_{t'}^c + \xi$. To quantify the distance between the generated local model $\theta_t^c$ and the Trojaned model $X$, we subtract $\theta_t^c$ by $X$ as follows: $\theta_t^c - X = (1 - \frac{1}{\psi_{t'}^c}) \triangle \bar{\theta}_{t'}^c + \xi$. Based upon this, we can bound the $l_2$-norm of the distance $\theta_t^c - X$ as follows:

$$\|\theta_t^c - X\|_2 = \|(1 - \frac{1}{\psi_{t'}^c}) \triangle \bar{\theta}_{t'}^c + \xi\|_2 \le (\frac{1}{a} - 1)\| \triangle \bar{\theta}_{t'}^c \|_2 + \|\xi\|_2 \tag{6}$$

Consequently, Theorem 1 holds. □

## 4 Experimental Results

We focus on answering the following three questions in our evaluation: **(1)** Whether HNTROJ is effective in HyperNetFL? **(2)** What is the percentage of compromised clients required for an effective attack? and **(3)** How difficult it is to defend against HNTROJ?

**Data and Model Configuration.** We conduct an extensive experiment on CIFAR-10 [21] and Fashion MNIST datasets [22]. For both datasets, we have 100 clients in which the data is non-iid

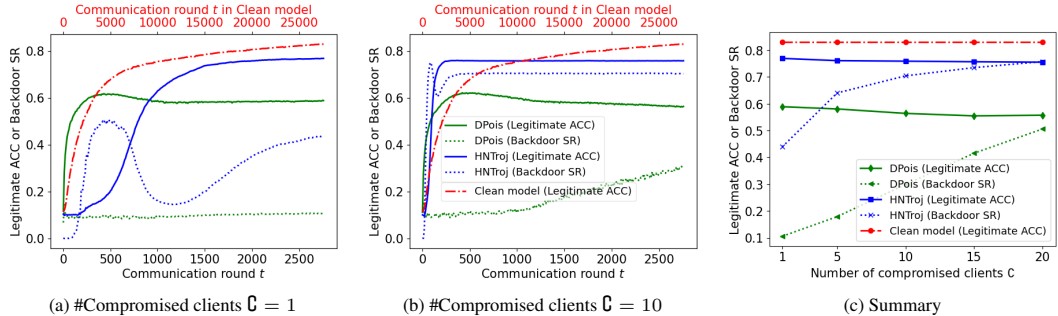

(a) #Compromised clients $\mathsf{C} = 1$      (b) #Compromised clients $\mathsf{C} = 10$      (c) Summary

Figure 2: Legitimate ACC and Backdoor SR comparison for DPOIS, HNTROJ, and Clean model over different numbers of compromised clients in the CIFAR-10 dataset. (Fig. 2a has the same legend as in Fig. 2b).

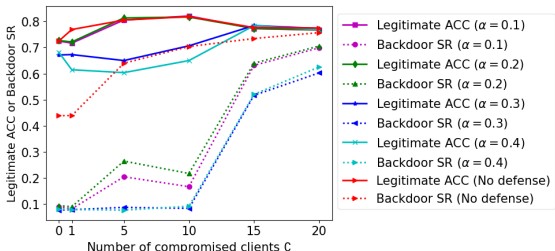

Figure 3: Legitimate ACC and backdoor SR under $\alpha$-trimmed norm defense in the CIFAR-10 dataset.

across clients and use the class 0 as a targeted class $y_j^b$. We adopt the model configuration described in [4]. We use WaNet [15] for generating backdoor data to train $X$. More information is in Appx. D.

**Evaluation Metrics.** For evaluation, we use:

$$\text{Legitimate ACC} = \frac{1}{N} \sum_{i \in [N]} \frac{1}{n_i^\tau} \sum_{j \in [n_i^\tau]} Acc\big(f(x_j^i, \theta^i), y_j^i\big)$$

$$\text{Backdoor SR} = \frac{1}{N} \sum_{i \in [N]} \frac{1}{n_i^\tau} \sum_{j \in [n_j^\tau]} Acc\big(f(\overline{x}_j^i, \theta^i), y_j^{i,b}\big)$$

where $\overline{x}_j^i = x_j^i + \mathcal{T}$ is a Trojaned sample, $Acc(y', y) = 1$ if $y' = y$; otherwise $Acc(y', y) = 0$ and $n_i^\tau$ is the number of testing samples in client $i$.

**HNTROJ v.s. DPOIS and White-box HNREPL.** Figs. 2 and 10a (Appx. D) present legitimate ACC and backdoor SR of each attack and the clean model (i.e., trained without poisoning) as a function of the communication round $t$ and the number of compromised clients $\mathsf{C}$ under a defense free environment in the CIFAR-10 dataset. It is obvious that HNTROJ significantly outperforms DPOIS. HNTROJ requires a notably small number of compromised clients to successfully backdoor the HyperNetFL with high backdoor SR, i.e., $43.92\%$, $64.00\%$, $70.38\%$, $73.45\%$, and $75.70\%$ compared with $10.58\%$, $17.87\%$, $29.90\%$, $41.53\%$, and $50.57\%$ of the DPOIS given 1, 5, 10, 15, and 20 compromised clients, without an undue cost in legitimate ACC, i.e., $76.89\%$.

In addition, HNTROJ does not introduce degradation or sudden shifts in legitimate ACC during the training process, regardless of the number of compromised clients, making it stealthier than DPOIS and (white-box) HNREPL. This is because we consistently poison the HyperNetFL with a relatively good model $X$, which achieves $74.26\%$ legitimate ACC and $85.92\%$ backdoor SR, addressing the inconsistency in deriving the malicious local gradients. There is a small legitimate ACC gap between HNTROJ and the clean model, i.e., $6.11\%$ in average. However, this gap will not be noticed by the server since the clean model is invisible to the server when the compromised clients are present.

**HNTROJ v.s. $\alpha$-Trimmed Norm.** Since HNTROJ outperforms other poisoning attacks, we now focus on understanding its performance under robust HyperNetFL training. Fig. 3 shows the performance of $\alpha$-trimmed norm against HNTROJ as a function of the number of compromised clients $\mathsf{C}$. There

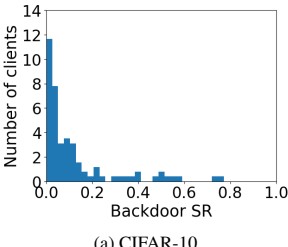 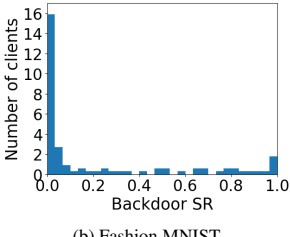

(a) CIFAR-10          (b) Fashion MNIST

Figure 4: Backdoor Success Rate at the Client Level.

are three key observations from the results, as follows: **(1)** Applying $\alpha$-trimmed norm does reduce the backdoor SR, especially when the number of compromised clients is small, i.e., backdoor SR drops $39.72\%$ in average given $\mathsf{C} \in [1, 10]$. However, when the number of compromised clients is a little bit larger, the backdoor SR is still at highly severe levels, i.e., $51.70\% \sim 70.55\%$ given $15$ to $20$ compromised clients, regardless of a wide range of trimming level $\alpha \in [0.1, 0.4]$; **(2)** The larger the $\alpha$ is, the lower the backdoor SR tends to be. This good result comes with a toll on the legitimate ACC, which is notably reduced when $\alpha$ is larger. In average, the legitimate ACC drops from $79.75\%$ to $67.66\%$ and $62.29\%$ given $\alpha \in [0.3, 0.4]$ and $\mathsf{C} \in [1, 10]$, respectively. That clearly highlights a non-trivial trade-off between legitimate ACC and backdoor SR given attacks and defenses; and **(3)** The more compromised clients we have, the better the legitimate ACC is when the trimming level $\alpha$ is large, i.e., $\alpha \in [0.3, 0.4]$. That is because training with the Trojaned model $X$, which has a relatively good legitimate ACC, can mitigate the damage of large trimming levels on the legitimate ACC. In fact, a large number of compromised clients implies a better probability for the compromised clients to sneak through the trimming; thus, improving both legitimate ACC and backdoor SR.

We observe a similar phenomenon when we apply the client-level DP optimizer as a defense against HNTROJ (Fig. 13). More information is in Appx. D.

**Backdoor SR at Client Level.** Importantly, the histogram of backdoor SR in the CIFAR-10 dataset under DP optimizer (Fig. 4a) shows that HNTROJ with only 1 compromised clients can open backdoors to 10 (a decent number of) legitimate clients with high backdoor SRs ($> 0.4$). We observe similar backdoor SRs to 22 legitimate clients with only 1 compromised client in FMNIST (Fig. 4b). Therefore, it is not easy to defend against HNTROJ.

**Results on the Fashion MNIST dataset.** The results on the Fashion MNIST dataset further strengthen our observation. DPOIS even failed to implant backdoors into HyperNetFL (Figs. 10b and 11, Appx. D). This is because the HyperNetFL model converges $10\mathbf{x}$ faster than the model for the CIFAR-10 dataset, i.e., given the simplicity of the Fashion MNIST dataset; thus, significantly reducing the poisoning probability through participating in the training of a small set of compromised clients. Technically, we found that the client-level DP optimizer appears having the potential to mitigate HNTROJ due to the model's fast convergence. However, the backdoor SR at the client level shows that with only 1 compromised client, HNTROJ can open backdoors to 22 (over 100) legitimate clients with SR $> 0.4$ (Fig. 4). Therefore, it is challenging for the client-level DP optimizer to defend HNTROJ. In addition, the $\alpha$-trimmed norm is still failed to defend again HNTROJ (Figs. 7, 12, and 14, Appx. D).

## 5   Conclusion and Future Work

We presented a black-box model transferring attack (HNTROJ) to implant backdoor into HyperNetFL. We overcome the lack of consistency in deriving malicious local gradients to efficiently transfer a Trojaned model to the outputs of the HyperNetFL. We multiply a random and dynamic learning rate to the malicious local gradients making the attack stealthy. To defend against HNTROJ, we adapted several robust FL training algorithms into HyperNetFL. Extensive experiment results show that HNTROJ outperforms black-box DPOIS and white-box HNREPL bypassing adapted robust training algorithms with small numbers of compromised clients. Future work is to 1) adapt HNTROJ on other personalized FL frameworks and 2) use multiple Trojaned models adapting to diverse compromised clients.

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

# A  Background and Related Work

## A.1  Federated Learning (FL)

We consider the following FL protocol: at round $t$, the server sends the latest model weights $\theta_t$ to a randomly sampled subset of clients $S_t$. Upon receiving $\theta_t$, the client $i$ in $S_t$ uses $\theta_t$ to train their local model for some number of iterations, e.g., via stochastic gradient descent (SGD), and results in model weights $\theta_t^i$. The client $i$ computes their local gradient $\triangle\theta_t^i = \theta_t^i - \theta_t$, and sends it back to the server. After receiving all the local gradients from all the clients in $S_t$, the server updates the model weights by aggregating all the local gradients by using an aggregation function $\mathcal{G} : R^{|S_t|\times m} \rightarrow R^m$ where $m$ is the size of $\triangle\theta_t^i$. The aggregated gradient will be added to $\theta_t$, that is, $\theta_{t+1} = \theta_t + \lambda\mathcal{G}(\{\triangle\theta_t^i\}_{i\in S_t})$ where $\lambda$ is the server's learning rate. A typical aggregation function is Federated Averaging (FedAvg) applied in many papers in FL [5], as follows:

$$\theta_{t+1} = \theta_t + \lambda(\sum_{i\in S_t} n_i \times \triangle\theta_t^i)/\sum_{i\in S_t} n_i \tag{7}$$

When the number of training samples $n_i$ is hidden from the server, one can use an unweighted aggregation function: $\theta_{t+1} = \theta_t + \lambda\sum_{i\in S_t}\triangle\theta_t^i/|S_t|$.

## A.2  Training protocol of a HyperNetFL

The training protocol of a HyperNetFL (Alg. 2) is generally similar to the aforementioned protocol of typical FL. However, at round $t$, there are three differences in training HyperNetFL compared with FedAvg [5]: **(1)** There is no global model $\theta$ generated by the aggregation function $\mathcal{G}$ in Eq. 7; **(2)** Each client $i \in S_t$ receives the personalized weights $\theta_t^i$ from the HyperNetFL. The client $i$ computes the local gradient $\triangle\theta_t^i$ and then sends it to the server; **(3)** The server uses all the local gradients received from all the clients $i \in S_t$ to update $\varphi$ and the descriptors $v_i$ using general update rules in Lines 9 and 10 (Alg. 2); and **(4)** The size of the HyperNetFL weights $\varphi$ is significanlty larger than $\theta$ ($\varphi \simeq 10$x size of $\theta$) causing extra computational cost at the server. This protocol is more general than using only one client at a communication round as in [4]. By using a small batch of clients per communication round, we observe that we can enhance the performance of the HyperNetFL and make it more reliable.

# B  Data Poisoning in HyperNetFL

We consider both white-box and black-box model replacement threat models. Although unrealistic, the white-box setting provides the upper bound risk. Meanwhile, the black-box setting will inform a realistic risk in practice. Details regarding the white-box setting and the adaptation of model replacement attacks into HyperNetFL, called **HNREPL**, are available[1]. Let us present our black-box threat model and attacks as follows.

**Black-box Threat Model.** At round $t$, an adversary fully controls a small set of compromised clients $\mathsf{C}$. The adversary cannot modify the training protocol of the HyperNetFL at the server and at the legitimate clients. The adversary's goal is to implant backdoors in all local models $\{\theta^i = h(v_i, \varphi)\}_{i\in[N]}$ by minimizing a backdoor objective:

$$\arg\min_{\varphi,\{v_i\}_{i\in[N]}} \frac{1}{N}\sum_{i=1}^{N}\left[L_i(h(v_i,\varphi)) + L_i^b(h(v_i,\varphi))\right] \tag{8}$$

---

[1]https://www.dropbox.com/s/cn3omnqx7sso6my/HNTroj-2.pdf?dl=0

**Algorithm 2** HyperNetFL with Multiple Clients per Round

---

**Input:** Number of rounds $T$, number of local rounds $K$, server's learning rates $\lambda$ and $\zeta$, clients' learning rate $\eta$, number of clients $N$, and $L_i(B)$ is the loss function $L_i(\theta)$ on a mini-batch $B$

**Output:** $\varphi, v_i$

1: **for** $t = 1, \ldots, T$ **do**
2:     Sample clients $S_t$
3:     **for** each client $i \in S_t$ **do**
4:         set $\theta_t^i = h(v_i, \varphi)$ and $\tilde{\theta}^i = \theta_t^i$
5:         **for** $k = 1, \ldots, K$ **do**
6:             sample mini-batch $B \subset D_i$
7:             $\tilde{\theta}_{k+1}^i = \tilde{\theta}_k^i - \eta \nabla_{\tilde{\theta}_k^i} L_i(B)$
8:         $\triangle\theta_t^i = \tilde{\theta}_K^i - \theta_t^i$
9:     $\varphi = \varphi - \frac{\lambda}{|S_t|} \sum_{i=1}^{|S_t|} (\nabla_\varphi \theta_t^i)^\top \triangle\theta_t^i$
10:     $\forall i \in S_t : v_i = v_i - \zeta \nabla_{v_i} \varphi^\top (\nabla_\varphi \theta_t^i)^\top \triangle\theta_t^i$

---

where $L_i^b$ is the (backdoor) loss function of the $i^{th}$ client given Trojaned examples $x_j + \mathcal{T}$ with the trigger $\mathcal{T}$ [15], e.g., $L_i^b = \frac{1}{n_i} \sum_{j=1}^{n_i} L(f(x_j + \mathcal{T}, \theta), y_j^b)$ where $y_j^b$ is the targeted label for the sample $x_j + \mathcal{T}$. One can vary the portion of Trojaned samples to optimize the attack performance. This black-box threat model is applied throughout this paper.

We found that HNREPL is infeasible in the black-box setting, since the weights $\varphi$ and the descriptors $\{v_i\}_{i \in [N]}$ are hidden from all the clients. Also, there is lack of effective approach to infer (large) $\varphi$ and $\{v_i\}_{i \in [N]}$ given a small number of compromised clients.

**Data Poisoning (DPOIS) in HyperNetFL.** To address the issues of HNREPL, we look into another fundamental approach, that is applying black-box DPOIS. The pseudo-code of the attack is in Alg. 3. At round $t$, the compromised clients $c \in \complement \cap S_t$ receive the personalized model weights $\theta_t^c$ from the server. Then, they compute malicious local gradients $\triangle\bar{\theta}_t^c$ using their Trojan datasets, i.e., their legitimate data combined with Trojaned data samples, to minimize their local backdoor loss functions: $\forall c \in \complement \cap S_t : \theta_t^{c*} = \arg\min_{\theta_t^c} [L_c(h(v_c, \varphi)) + L_c^b(h(v_c, \varphi))]$, after a certain number of local steps $K$ of SGD. All the malicious local gradients $\{\triangle\bar{\theta}_t^c\}_{c \in \complement}$ are sent to the server. If the HyperNetFL updates the model weights $\varphi$ and the descriptors $\{v_i\}_{i \in S_t}$ using $\{\triangle\bar{\theta}_t^c\}_{c \in \complement}$, the local model weights generated by the HyperNetFL $h(\cdot, \varphi)$ will be Trojan infected. This is because the update rules of the HyperNetFL become the gradient of an approximation to the following Trojaned surrogate loss

$$\frac{1}{2}\Big(\sum_{c \in \complement} \|\theta^{c*} - h(v_c, \varphi)\|_2^2 + \sum_{i \in N \setminus \complement} \|\theta^{i*} - h(v_i, \varphi)\|_2^2\Big) \tag{9}$$

where $\theta^{c*}$ is the optimal local Trojaned model weights, $i \in N \setminus \complement$ are legitimate clients and their associated legitimate loss functions $\theta^{i*} = \arg\min_{\theta^i} L_i(h(v_i, \varphi))$.

**Disadvantages of DPOIS.** Obviously, the larger the number of compromised clients is, i.e., a larger $\complement$ and a smaller $N \setminus \complement$, the more effective the attack will be. Although more practical than the HNREPL in poisoning HyperNetFL, there are two issues in the black-box DPOIS: **(1)** The attack causes notable degradation in model utility on the legitimate data samples; and **(2)** The attack requires a more significant number of compromised clients to be successful. These disadvantages reduce the stealthiness and effectiveness of the attack, respectively.

The root cause issue of the DPOIS is the lack of consistency in deriving the malicious local gradient $\triangle\bar{\theta}_t^c = \theta_t^{c*} - h(v_c, \varphi)$ across communication rounds and among compromised clients to outweigh the local gradients from legitimate clients. First, $\theta_t^{c*}$ is derived after (a small number) $K$ local steps of applying SGD to minimize the local backdoor loss function $\theta_t^{c*} = \arg\min_{\theta_t^c} [L_c(h(v_c, \varphi)) + L_c^b(h(v_c, \varphi))]$, in which the local model weights $h(v_c, \varphi)$ and the loss functions (i.e, $L_c$ and $L_c^b$) are varying among compromised clients due to the descriptors $\{v_c\}_{c \in \complement}$ in addition to the their dissimilar local datasets. As a result, the (supposed to be) Trojaned model weights $\{\theta_t^{c*}\}_{c \in \complement}$ are unlike among compromised clients. Second, a small number of local training steps $K$ (i.e., given a limited computational power on the compromised clients) is not sufficient to approximate a good Trojaned model $\theta_t^{c*}$. The adversary can increase the local training steps $K$ if more computational power is available. However, there is still no guarantee that $\{\theta_t^{c*}\}_{c \in \complement}$ will be alike without the

---

**Algorithm 3** Backdoor Data Poisoning in HyperNetFL (DPOIS)

---

**Input:** Number of rounds $T$, number of local rounds $K$, server's learning rates $\lambda$ and $\zeta$, clients' learning rate $\eta$, number of clients $N$, and $L_c(B)$ and $L_c^b(B)$ are the loss functions (legitimate) $L_c(\theta)$ and (backdoor) $L_c^b(\theta)$ on a mini-batch $B$

**Output:** $\varphi, v_i$

1:  **for** $t = 1, \dots, T$ **do**
2:      Sample clients $S_t$
3:      **for** each legitimate client $i \in S_t \setminus \complement$ **do**
4:          set $\theta_t^i = h(v_i, \varphi)$ and $\tilde{\theta}^i = \theta_t^i$
5:          **for** $k = 1, \dots, K$ **do**
6:              sample mini-batch $B \subset D_i$
7:              $\tilde{\theta}_{k+1}^i = \tilde{\theta}_k^i - \eta \nabla_{\tilde{\theta}_k^i} L_i(B)$
8:          $\triangle \theta_t^i = \tilde{\theta}_K^i - \theta_t^i$
9:      **for** each compromised client $c \in S_t \cap \complement$ **do**
10:         set $\theta_t^c = h(v_c, \varphi)$ and $\tilde{\theta}^c = \theta_t^c$
11:         **for** $k = 1, \dots, K$ **do**
12:             sample mini-batch $B \subset D_c$
13:             $\tilde{\theta}_{k+1}^c = \tilde{\theta}_k^c - \eta \nabla_{\tilde{\theta}_k^c} [L_c(B) + L_c^b(B)]$
14:         $\triangle \bar{\theta}_t^c = \tilde{\theta}_K^c - \theta_t^c$
15:     $\varphi = \varphi - \frac{\lambda}{|S_t|} \sum_{i=1}^{|S_t|} (\nabla_\varphi \theta_t^i)^\top \triangle \theta_t^i$
16:     $\forall i \in S_t : v_i = v_i - \zeta \nabla_{v_i} \varphi^\top (\nabla_\varphi \theta_t^i)^\top \triangle \theta_t^i$

---

control over the dissimilar descriptors $\{v_c\}_{c \in \complement}$. Third, the local model weights $h(v_c, \varphi)$ change after every communication round and are heavily affected by the local gradients from legitimate clients. Consequently, the malicious local gradients $\triangle \bar{\theta}_t^c$ derived across all the compromised clients $c \in \complement$ do not synergistically optimize the approximation to the Trojaned surrogate loss function (Eq. 9) such that the outputs of the HyperNetFL $h(\cdot, \varphi)$ are Trojan infected.

Therefore, developing a practical, stealthy, and effective backdoor attack in HyperNetFL is non-trivial and remains an open problem.

## C   Robust HyperNetFL Training

In this section, we first investigate the state-of-the-art defenses against backdoor poisoning in FL and point out the differences between FL and HyperNetFL. We then present our robust training approaches adapted from existing defenses for HyperNetFL against HNTROJ.

Existing defense approaches against backdoor poisoning in ML can be categorised into two lines: 1) Trojan detection in the inference phase and 2) robust aggregation to mitigate the impacts of malicious local gradients in aggregation functions. In this paper, we applied the state-of-the-art warping-based Trojans bypassing all the well-known Trojan detection methods, i.e., Neural Cleanse [16], Fine-Pruning [16], and STRIP [18], in the inference phase. HNTROJ does not affect the warping-based Trojans (Fig. 9); thus bypassing these detection methods. Based upon that, we focus on identifying which robust aggregation approaches can be adapted to HyperNetFL and how.

**Robust Aggregation** Several works have proposed robust aggregation approaches to deter byzantine attacks in typical FL, such as coordinate-wise median, geometric median, $\alpha$-trimmed mean, or a variant and combination of such techniques [23]. Recent approaches include weight-clipping and noise addition with certified bounds, ensemble models, differential privacy (DP) optimizers, and adaptive and robust learning rates (RLR) across clients and at the server [12, 24].

Despite differences, existing robust aggregation focuses on analysing and manipulating the local gradients $\triangle \theta_t^i$, which share the global aggregated model $\theta_t$ as the same root, i.e., $\forall i \in [N] : \triangle \theta_t^i = \theta_t^i - \theta_t$. The fundamental assumption in these approaches is that the local gradients from compromised clients $\{\triangle \bar{\theta}_t^c\}_{c \in \complement}$ and legitimate clients $\{\triangle \theta_t^i\}_{i \in N \setminus \complement}$ are different in terms of magnitude and direction.

**Robust FL Training v.s. HyperNetFL.** Departing from typical FL, the local gradients in HyperNetFL have different and personalized roots $h(v_i, \varphi)$, i.e., $\forall i \in [N] : \triangle \theta_t^i = \theta_t^{i*} - h(v_i, \varphi)$. Therefore, the

---

**Algorithm 4** Client-level DP Optimizer in HyperNetFL

---

**Input:** Number of rounds $T$, number of local rounds $K$, server's learning rates $\lambda$ and $\zeta$, clients' learning rate $\eta$, number of clients $N$, clipping bound $\mu$, noise scale $\sigma$, and $L_i(B)$ is the loss function $L_i(\theta)$ on a mini-batch $B$

**Output:** $\varphi, v_i$

1:   **for** $t = 1, \dots, T$ **do**
2:       Sample clients $S_t$
3:       **for** each client $i \in S_t$ **do**
4:           set $\theta_t^i = h(v_i, \varphi)$ and $\tilde{\theta}^i = \theta_t^i$
5:           **for** $k = 1, \dots, K$ **do**
6:               sample mini-batch $B \subset D_i$
7:               $\tilde{\theta}_{k+1}^i = \tilde{\theta}_k^i - \eta \bigtriangledown_{\tilde{\theta}_k^i} L_i(B)$
8:           $\triangle\theta_t^i = \tilde{\theta}_K^i - \theta_t^i$
9:       $\varphi = \varphi - \frac{\lambda}{|S_t|} \Big[ \sum_{i=1}^{|S_t|} \frac{(\bigtriangledown_\varphi \theta_t^i)^\top \triangle\theta_t^i}{\max(1, \frac{\|(\bigtriangledown_\varphi \theta_t^i)^\top \triangle\theta_t^i\|_2}{\mu})} + \mathcal{N}(0, \sigma^2\mu^2\mathbf{I}) \Big]$
10:      $\forall i \in S_t : v_i = v_i - \zeta \bigtriangledown_{v_i} \varphi^\top \frac{(\bigtriangledown_\varphi \theta_t^i)^\top \triangle\theta_t^i}{\max(1, \frac{\|(\bigtriangledown_\varphi \theta_t^i)^\top \triangle\theta_t^i\|_2}{\mu})}$

---

local gradients $\{\triangle\theta_t^i\}_{i \in N}$ in HyperNetFL may diverge in magnitude and direction in their own sub-optimal spaces, making it challenging to adapt existing robust aggregation methods into HyperNetFL. More importantly, manipulating the local gradients alone can significantly affect the original update rules of the HyperNetFL, which are derived based on the combination between the local gradients and the derivatives of $\varphi$ and $v_i$ given the output of $h(\cdot, \varphi)$, i.e., $(\bigtriangledown_\varphi \theta_t^i)^\top$ and $\bigtriangledown_{v_i} \varphi^\top (\bigtriangledown_\varphi \theta_t^i)^\top$, respectively.

For instance, adapting the recently developed RLR [12] on the local gradients can degrade the model utility on legitimate data samples to a random guess level on several benchmark datasets[1]. In addition, the significantly large size of $\varphi$ introduces an expensive computational cost in adapting (statistics-based) robust aggregation approaches into HyperNetFL against HNTROJ.

Regarding certified bounds against backdoors at the inference phase derived in weight-clipping and noise addition [25] cannot be straightforwardly adapted to HyperNetFL, since the bounds are especially designed for the aggregated model $\theta$ (e.g., Eq. 7). There is no such aggregated model in HyperNetFL. Similarly, the ensemble model training-based robustness bounds [26] cannot be directly adapted to HyperNetFL, since there is no ensemble model training in HyperNetFL.

**Robust HyperNetFL Training.** Based on our observation, to avoid damaging the update rule of HyperNetFL, a suitable way to develop robust HyperNetFL training algorithms is to adapt existing robust aggregation on the set of $\varphi$'s gradients $\{(\bigtriangledown_\varphi \theta_t^i)^\top \triangle\theta_t^i\}_{i \in S_t}$ given $\varphi = \varphi - \frac{\lambda}{|S_t|}\sum_{i=1}^{|S_t|}(\bigtriangledown_\varphi \theta_t^i)^\top \triangle\theta_t^i$. It is worth noting that we may not need to modify the update rule of the descriptors $\{v_i\}_{i \in S_t}$ since $v_i$ is a personalized update that does not affect the updates of any other descriptors $\{v_j\}_{j \neq i}$ and the model weights $\varphi$.

**Client-level DP Optimizer.** Among robust training against backdoor poisoning attacks, differential privacy (DP) optimizers, weight-clipping and noise addition can be adapted to defend against HNTROJ. Since they share the same spirit, that is, clipping gradients from all the clients before adding noise into their aggregation, we only consider DP optimizers in this paper without loss of generality. In specific, we consider a DP optimizer, which can be understood as the weight updates $\varphi$ are not excessively influenced by any of the local gradients $(\bigtriangledown_\varphi \theta_t^i)^\top \triangle\theta_t^i$ where $i \in S_t$. By clipping every $\varphi$'s gradients under a pre-defined $l_2$-norm $\mu$, we can bound the influence of a single client's gradient $(\bigtriangledown_\varphi \theta_t^i)^\top \triangle\theta_t^i$ to the model weights $\varphi$. To make the gradients indistinguishable, we add Gaussian noise into the $\varphi$'s gradient aggregation: $\varphi = \varphi - \frac{\lambda}{|S_t|}\Big[ \sum_{i=1}^{|S_t|}(\bigtriangledown_\varphi \theta_t^i)^\top \triangle\theta_t^i + \mathcal{N}(0, \sigma^2\mu^2\mathbf{I})\Big]$, where $\sigma$ is a predefined noise scale. The pseudo-code of our approach is in Alg. 4. We utilize this client-level DP optimizer as an effective defense against HNTROJ. As in [24], we focus on how parameter configurations of the client-level DP optimizer defend against HNTROJ with minimal utility loss, regardless of the privacy provided.

$\alpha$-**Trimmed Norm** In addition, among robust aggregation approaches against byzantine attacks, median-based approaches, $\alpha$-trimmed mean, and variants of these techniques [27, 28] can be adapted to HyperNetFL against HNTROJ by applying them on the gradients of $\varphi$. Without loss of generality, we adapt the well-applied $\alpha$-trimmed mean approach [23] into HyperNetFL to eliminate potentially malicious $\varphi$'s gradients in this paper. The adapted algorithm needs to be less computational resource hungry in order for it to efficiently work with the large size of $\varphi$. Therefore, instead of looking into each element of the $\varphi$'s gradient as in $\alpha$-trimmed mean, we trim the top $\frac{\alpha}{2}\%$ and the bottom $\frac{\alpha}{2}\%$ of the gradients $\{(\nabla_\varphi \theta_t^i)^\top \triangle \theta_t^i\}_{i \in S_t}$ that respectively have the highest and lowest magnitudes quantified by an $l_2$-norm, i.e., $\|(\nabla_\varphi \theta_t^i)^\top \triangle \theta_t^i\|_2$. The remaining gradients after the trimming, denoted $\{(\nabla_\varphi \theta_t^i)^\top \triangle \theta_t^i\}_{i \in S_t^{\alpha-trim}}$, are used to update the HyperNetFL model weights $\varphi$, i.e., $\varphi = \varphi - \frac{\lambda}{|S_t^{\alpha-trim}|} \sum_{i=1}^{|S_t^{\alpha-trim}|} (\nabla_\varphi \theta_t^i)^\top \triangle \theta_t^i$. The descriptors $\{v_i\}_{i \in S_t^{\alpha-trim}}$ are updated normally. The pseudo-code of the $\alpha$-trimmed norm for HyperNetFL is in Alg. 5.

# D  Experiments

**Data and Model Configuration.** To achieve our goal, we conduct an extensive experiment on CIFAR-10 [21] and Fashion MNIST datasets [22]. For both datasets, to generate non-iid data distribution across clients in terms of classes and size of local training data, we randomly sample two classes for each client. And for each client $i$ and selected class $c$, we sample $p_{i,c} \sim \mathcal{N}(0, 1)$ and assign it with $\frac{p_{i,c}}{\sum_j p_{j,c}}$ of the samples for this class. We use 100 clients. There are $60,000$ samples in the CIFAR-10 and $70,000$ samples in the Fashion MNIST datasets. Each dataset is divided into three non-overlapping sets: $10,000$ samples for testing, $10,000$ samples for training the Trojaned model $X$, and the rest for training (i.e., $40,000$ samples in the CIFAR-10 and $50,000$ samples in the Fashion MNIST for training). We use the class 0 as a targeted class $y_j^b$ (Eq. 8) in each dataset.

We adopt the model configuration described in [4] for both datasets, in which we use a LeNet-based network [29] with two convolution and two fully connected layers for the local model and a fully-connected network with three hidden layers and multiple linear heads per target weight tensor for the HyperNetFL. SGD optimizer with the learning rate 0.01 for the HyperNetFL and 0.001 for the local model are used.

We use WaNet [15], which is one of the state-of-the-art backdoor attacks, for generating backdoor data. WaNet uses image warping-based triggers making the modification in the backdoor images natural and unnoticeable to humans. We follow the learning setup described in [15] to generate the backdoor images that are used to train $X$. In the DP optimizer, we vary the noise scale $\sigma \in \{10^{-1}, 10^{-2}, 10^{-3}, 10^{-4}\}$ and the clipping $l_2$-norm $\mu \in \{8, 4, 2, 1\}$. For the $\alpha$-trimmed norm approach, we choose $\alpha \in \{0.1, 0.2, 0.3, 0.4\}$.

**Evaluation Approach.** We carry out the validation through three approaches. We first compare HNTROJ with DPOIS and HNREPL in terms of legitimate accuracy (ACC) on legitimate data samples and backdoor successful rate (SR) on Trojaned data samples with a wide range number of compromised clients. The second approach is to investigate the effectiveness of adapted robust HyperNetFL training algorithms, including the client-level DP optimizer and the $\alpha$-trimmed norm, under a variety of hyper-parameter settings against HNTROJ. Based upon that, the third approach provides a performance summary of both attacks and defenses to inform the surface of backdoor risks in HyperNetFL. The (average) legitimate ACC and backdoor SR across clients on testing data are as follows:

$$\text{Legitimate ACC} = \frac{1}{N} \sum_{i \in [N]} \frac{1}{n_i^\tau} \sum_{j \in [n_i^\tau]} Acc\big(f(x_j^i, \theta^i), y_j^i\big)$$

$$\text{Backdoor SR} = \frac{1}{N} \sum_{i \in [N]} \frac{1}{n_i^\tau} \sum_{j \in [n_j^\tau]} Acc\big(f(\overline{x}_j^i, \theta^i), y_j^{i,b}\big)$$

where $\overline{x}_j^i = x_j^i + \mathcal{T}$ is a Trojaned sample, $Acc(y', y) = 1$ if $y' = y$; otherwise $Acc(y', y) = 0$ and $n_i^\tau$ is the number of testing samples in client $i$.

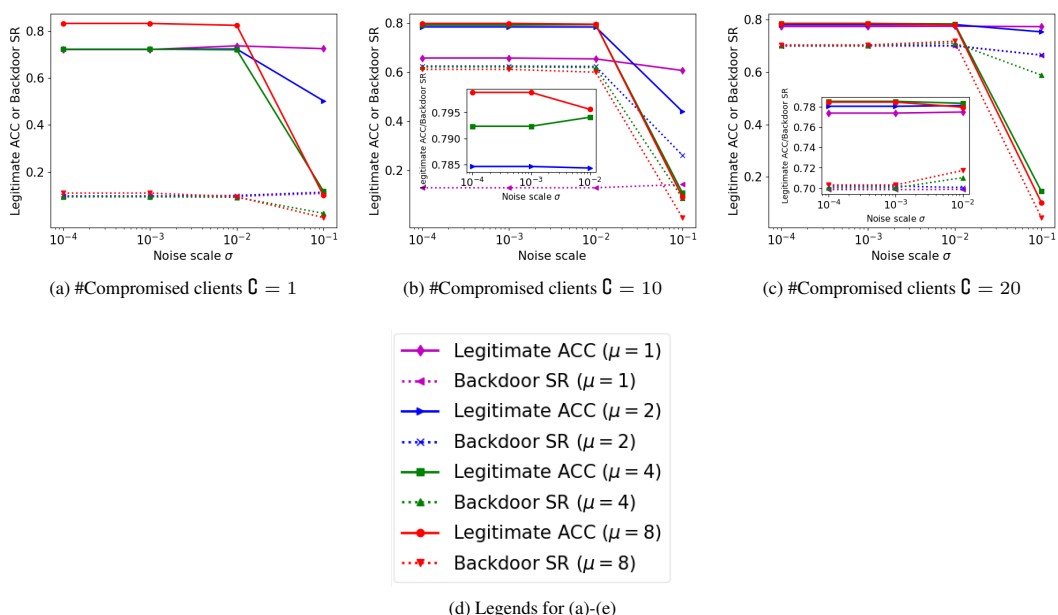

(d) Legends for (a)-(e)

Figure 5: HNTROJ under client-level DP optimizer-based robust HyperNetFL training in the CIFAR-10 dataset.

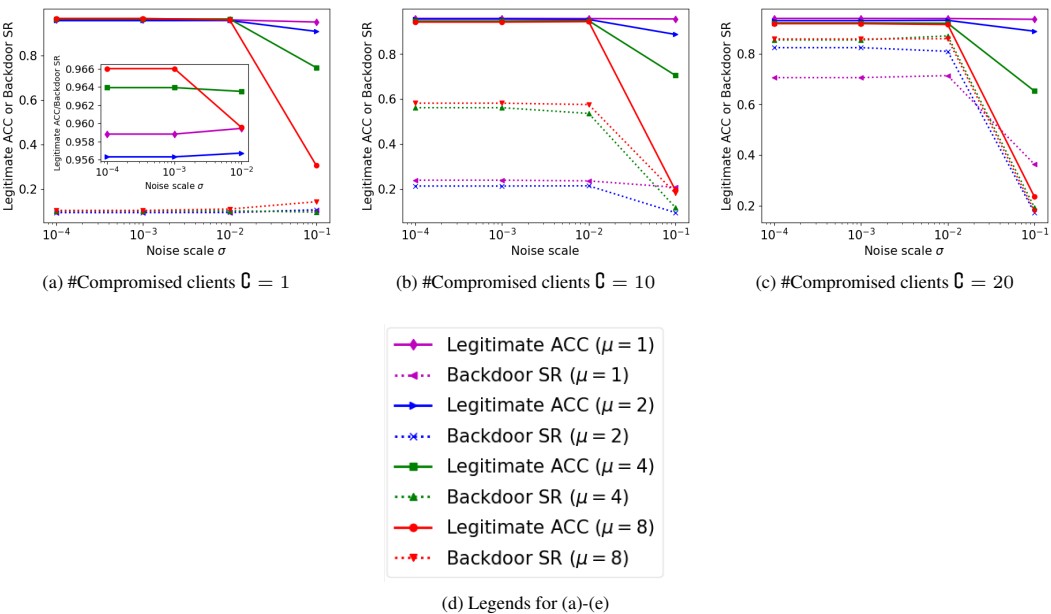

(d) Legends for (a)-(e)

Figure 6: HNTROJ under client-level DP optimizer-based robust HyperNetFL training in the Fashion MNIST dataset.

**Backdoor Risk Surface: Attacks and Defenses.** The trade-off between legitimate ACC and backdoor SR is non-trivially observable given many attack and defense configurations. To inform a better surface of backdoor risks, we look into a natural question: *"What can the adversary or the defender achieve given a specific number of compromised clients?"*

We answer this question by summarizing the best defending performance and the most stealthy and severe backdoor risk across hyper-parameter settings in the same diagram. Given a number of compromised clients $\complement$ and a robust training algorithm $\mathcal{A}$, the best defending performance, which maximizes both the 1) legitimate ACC (i.e., ACC in short) and 2) the gap between the ACC and

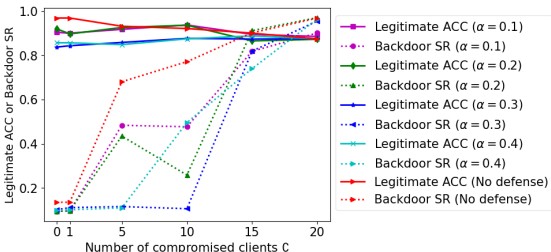

Figure 7: Legitimate ACC and backdoor SR under $\alpha$-trimmed norm defense in the Fashion MNIST dataset.

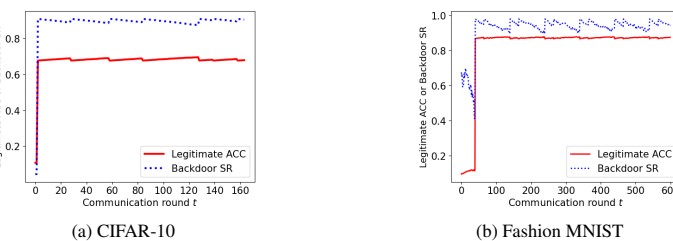

     (a) CIFAR-10              (b) Fashion MNIST

Figure 8: White-box model replacement attack in the CIFAR-10 and the Fashion MNIST datasets.

backdoor SR (i.e., SR in short), is identified across hyper-parameters' space of $\mathcal{A}$, i.e., $\Theta_{\mathcal{A}}^{\complement}$:

$$\phi^* = \arg\max_{\phi \in \Theta_{\mathcal{A}}^{\complement}} \left[ ACC(\phi) + \left( ACC(\phi) - SR(\phi) \right) \right]$$

where $ACC(\phi)$ and $SR(\phi)$ are the legitimate ACC and backdoor SR using the specific hyper-parameter configuration $\phi$. Similarly, we identify the most stealthy and severe backdoor risk maximizing both the legitimate ACC and backdoor SR:

$$\phi^* = \arg\max_{\phi \in \Theta_{\mathcal{A}}^{\complement}} \left[ ACC(\phi) + SR(\phi) \right]$$

Fig. 3 summaries the best performance of both defenses and attacks in the CIFAR-10 dataset as a function of the number of compromised clients through out the hyper-parameter space. For instance, given the number of compromised clients $\complement = 10$, using the client-level DP optimizer, the best defense can reduce the backdoor SR to $12.88\%$ with a cost of $13.45\%$ drop in the legitimate ACC. Meanwhile, in a weak defense using the client-level DP optimizer, the adversary can increase the backdoor SR up to $71.74\%$ without sacrificing much the legitimate ACC. From Figs. 3a-b, we can observe that $\alpha$-trimmed norm is a little bit more effective than the client-level DP optimizer by having a wider gap between legitimate ACC and backdoor SR. From the adversary angle, to ensure the success of the HNTROJ regardless of the defenses, the adversary needs to have at least $15$ compromised clients.

**HNTROJ v.s. Client-level DP Optimizer.** We observe a similar phenomenon when we apply the client-level DP optimizer as a defense against HNTROJ (Fig. 13). First, by using small noise scales $\sigma \in [10^{-4}, 10^{-2}]$, the client-level DP optimizer is effective in defending against HNTROJ when the number of compromised clients is small ($\complement \in [1, 5]$) achieving low backdoor SR, i.e., $31.97\%$ in average, while maintaining an acceptable legitimate ACC, i.e., $72.33\%$ in average. When the number of compromised clients is a little bit larger, the defense pays notably large tolls on the legitimate ACC (i.e., the legitimate ACC drops from $75.80\%$ to $31.18\%$ given $\complement = 10$) or fails to reduce the backdoor SR (i.e., backdoor SR $> 58.25\%$ given $\complement > 10$). That is consistent with our analysis. A small sufficient number of compromised clients synergistically and consistently can pull the outputs of the HyperNetFL to the Trojaned model $X$'s surrounded area.

**Algorithm 5** $\alpha$-Trimmed Norm in HyperNetFL

**Input:** Number of rounds $T$, number of local rounds $K$, server's learning rates $\lambda$ and $\zeta$, clients' learning rate $\eta$, number of clients $N$, and $L_i(B)$ is the loss function $L_i(\theta)$ on a mini-batch $B$

**Output:** $\varphi, v_i$

1:     **for** $t = 1, \ldots, T$ **do**
2:        Sample clients $S_t$
3:        **for** each client $i \in S_t$ **do**
4:           set $\theta_t^i = h(v_i, \varphi)$ and $\tilde{\theta}^i = \theta_t^i$
5:           **for** $k = 1, \ldots, K$ **do**
6:              sample mini-batch $B \subset D_i$
7:              $\tilde{\theta}_{k+1}^i = \tilde{\theta}_k^i - \eta \nabla_{\tilde{\theta}_k^i} L_i(B)$
8:           $\triangle \theta_t^i = \tilde{\theta}_K^i - \theta_t^i$
9:           $S_t^{\alpha-trim} = \alpha\text{-trimmed norm}\left(\{(\nabla_\varphi \theta_t^i)^\top \triangle \theta_t^i\}_{i \in S_t}\right)$
10:          $\varphi = \varphi - \frac{\lambda}{|S_t^{\alpha-trim}|} \sum_{i=1}^{|S_t^{\alpha-trim}|} (\nabla_\varphi \theta_t^i)^\top \triangle \theta_t^i$
11:          $\forall i \in S_t^{\alpha-trim} : v_i = v_i - \zeta \nabla_{v_i} \varphi^\top (\nabla_\varphi \theta_t^i)^\top \triangle \theta_t^i$

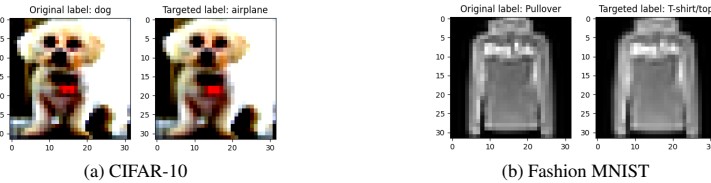

                 (a) CIFAR-10                             (b) Fashion MNIST

Figure 9: Legitimate samples (left) and their backdoor samples generated by WaNet [15] (right) in HNTROJ. They are are almost identical.

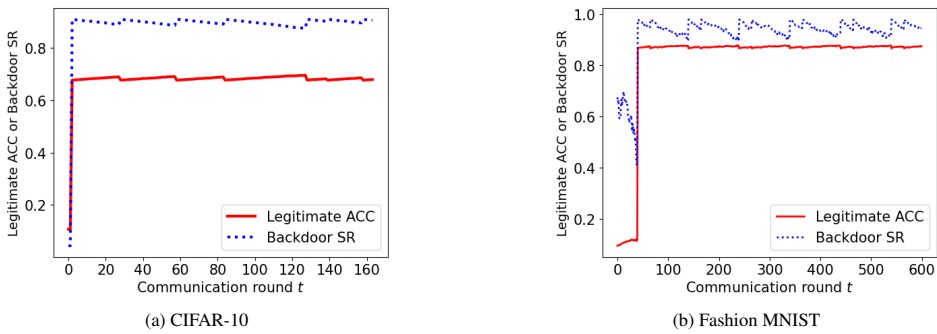

                 (a) CIFAR-10                             (b) Fashion MNIST

Figure 10: White-box model replacement attack in the CIFAR-10 and the Fashion MNIST datasets.

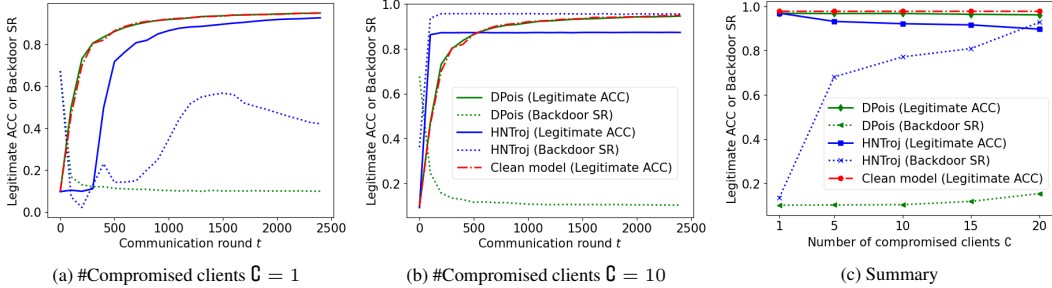

   (a) #Compromised clients $\mathtt{C} = 1$        (b) #Compromised clients $\mathtt{C} = 10$          (c) Summary

Figure 11: Legitimate ACC and Backdoor SR comparison for DPOIS, HNTROJ, and Clean model over different numbers of compromised clients in the Fashion MNIST dataset. (Fig. 11a and b have the same legend.)

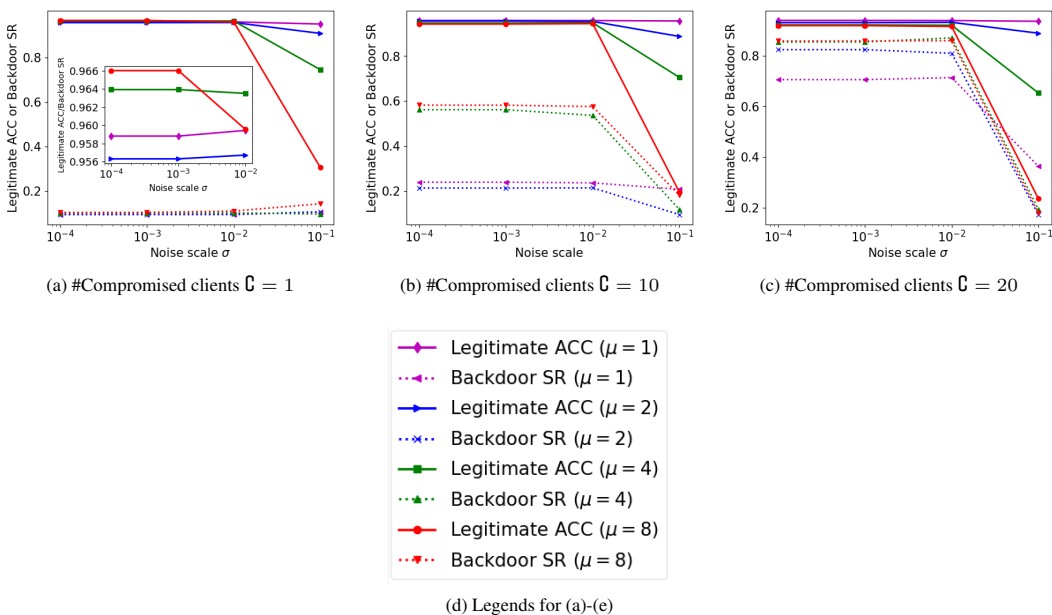

(a) #Compromised clients $C = 1$

(b) #Compromised clients $C = 10$

(c) #Compromised clients $C = 20$

(d) Legends for (a)-(e)

Figure 12: HNTROJ under client-level DP optimizer-based robust HyperNetFL training in the Fashion MNIST dataset.

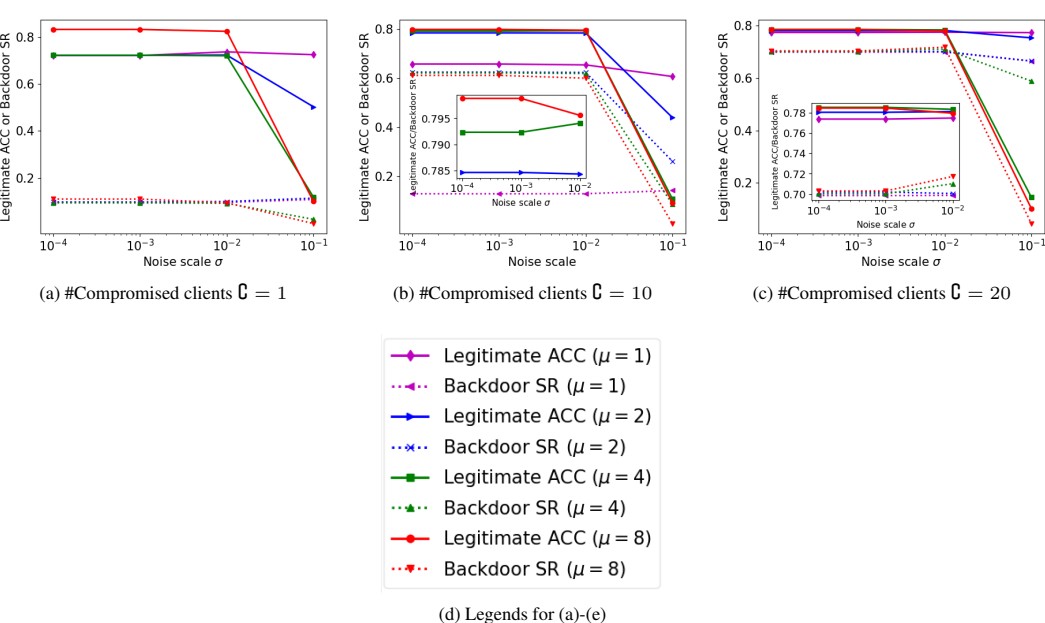

(a) #Compromised clients $C = 1$

(b) #Compromised clients $C = 10$

(c) #Compromised clients $C = 20$

(d) Legends for (a)-(e)

Figure 13: HNTROJ under client-level DP optimizer-based robust HyperNetFL training in the CIFAR-10 dataset.

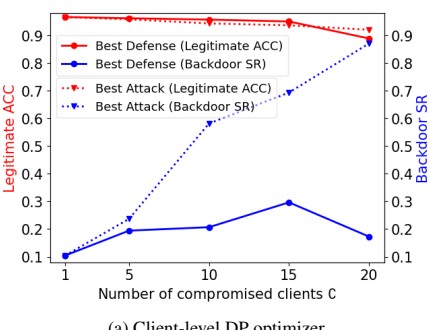
(a) Client-level DP optimizer

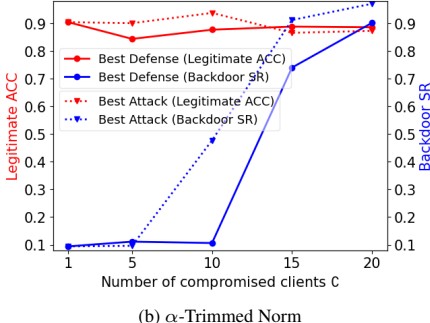
(b) $\alpha$-Trimmed Norm

Figure 14: Backdoor risk surface: attacks and defenses in the Fashion MNIST dataset. The attack we used here is HNTROJ.

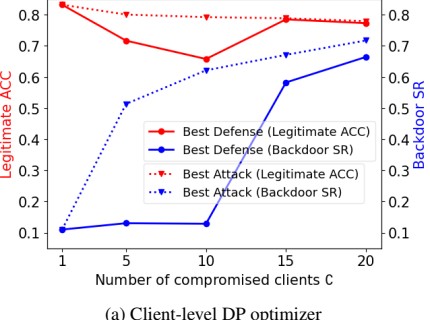
(a) Client-level DP optimizer

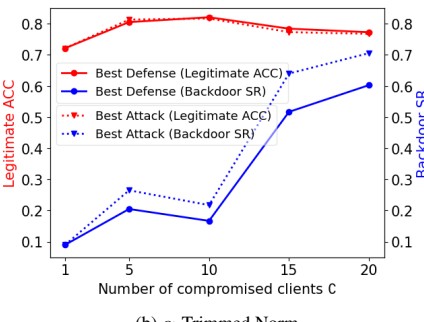
(b) $\alpha$-Trimmed Norm

Figure 15: Backdoor risk surface: attacks and defenses in the CIFAR-10 dataset. The attack we used here is HNTROJ.

