# OpenReview forum: "How to Backdoor HyperNetwork in Personalized Federated Learning?"
_NeurIPS.cc/2023/Workshop/BUGS — NeurIPS 2023 BUGS Poster_

### Official Review · Reviewer_rxwL · 2023-10-24
**A  simple but effective attack on HyperNetFL**

**Rating:** 6
**Confidence:** 4

**Review:**

This paper proposes a new attack on HyperNetFL where the server maintains a hypernetwork and generates a set of models from the hypernetwork for different clients for personalization. The authors propose a learning rate randomization strategy during the replacement of the clean weights with backdoored weights so as to make the replacement more stealthy. Theoretical analyses were given to show the minimum impact of the backdoored weights on the global model and its stealthiness. Extensive experiments prove the effectiveness of the proposed attack.

Weaknesses:
1. The abstract states "To defend against HNTROJ, we adapted several backdoor-resistant FL training algorithms into HyperNetFL", but none of these adapted defenses work. The authors should point out a direction for promising defense;

2. Some of the claims are hard to follow. Like on page 3 it says " This is because X has a better model utility on legitimate data sample", why? How does the adversary know for sure X is better?

3. In Alg. 1, line 10, why uniform [a, b]? Why not Bernoulli distribution？

4. How does the proposed attack work for standard FL?

---

### Official Review · Reviewer_9T3d · 2023-10-25
**Interesting scenario but needs improvement**

**Rating:** 6
**Confidence:** 3

**Review:**

This paper proposes a backdoor attack against HyperNet-based personalized federated learning. It aims to inject backdoors into all client models. The paper first generates a trojaned model and then uses its gradient to replace those of compromised clients. The gradients are obtained by multiplying a random value (between 0 and 1) with the trojaned model's gradient. The experiments on CIFAR-10 and Fashion MNIST datasets show that the proposed attack surpasses the baseline data poisoning attack.

Strengths

1. This paper targets a new federated learning scenario, which has not been studied in the literature.

2. Using one single trojaned model as the base for manipulating compromised clients is interesting.

Weaknesses

1. It is unclear why using the gradient from one single model does not affect the utility. This paper only says "X has a better model utility on legitimate data samples than the local models of legitimate clients." But there is no explanation for this. Actually, from Figure 2, there is non-trivial accuracy degradation compared to the clean model. It seems the proposed attack affects the utility.

2. The paper is hard to understand. It is not clear how HyperNet-based personalized federated learning works. The paragraph in Section 2 does not explain it clearly. What is "HNREPL"? There is no description of this abbreviation.

---

### Decision · Program_Chairs · 2023-10-28

**Decision:**

Accept (Poster)

**Comment:**

Thanks for submitting to BUGS workshop! Both reviewers recommend acceptance.